# Restoration of Sestrin 3 Expression Mitigates Cardiac Oxidative Damage in Ischemia–Reperfusion Injury Model

**DOI:** 10.3390/antiox14010061

**Published:** 2025-01-07

**Authors:** Mina Park, Sunghye Cho, Dongtak Jeong

**Affiliations:** Department of Medicinal & Life Science, College of Science and Convergence Technology, Hanyang University—ERICA, Ansan 15588, Republic of Korea; lampicka7119@hanyang.ac.kr (M.P.); josunghye88@gmail.com (S.C.)

**Keywords:** ischemia–reperfusion injury, sestrin, miR-25, ROS, oxidative damage, apoptosis, cardiac dysfunction, hypoxia/reoxygenation, antioxidant

## Abstract

Cardiac ischemia–reperfusion injury (IRI) occurs when blood flow is restored to the myocardium after a period of ischemia, leading to oxidative stress and subsequent myocardial cell damage, primarily due to the accumulation of reactive oxygen species (ROS). In our previous research, we identified that miR-25 is significantly overexpressed in pressure overload-induced heart failure, and its inhibition improves cardiac function by restoring the expression of SERCA2a, a key protein involved in calcium regulation. In this study, we aimed to investigate the role of miR-25 in the context of ischemia–reperfusion injury. We found that miR-25 was markedly upregulated under hypoxic conditions in both in vitro and in vivo models. Through in silico analysis, we identified Sestrin3 (SESN3), an antioxidant protein known for its protective effects against oxidative stress, as a novel target of miR-25. Based on these findings, we hypothesized that inhibiting miR-25 would restore Sestrin3 expression, thereby reducing ROS-induced myocardial cell damage and improving cardiac function. To test this hypothesis, we employed two model systems: a hypoxia/reoxygenation (H/R) stress model using H9c2 myoblasts and a surgically induced ischemia–reperfusion injury mouse model. Our results demonstrated that the use of miR-25 inhibitors significantly improved cardiac function and reduced myocardial damage in both models through the restoration of SESN3 expression. In conclusion, our findings suggest that targeting miR-25 may serve as a novel therapeutic modality to alleviate oxidative damage in the heart.

## 1. Introduction

Ischemic heart disease progresses to heart failure through myocardial infarction, resulting in negative outcomes [1,2,3]. A significant contributor to these results is oxidative stress, which arises from the sudden increase in oxygen supply when blood flow is restored to the ischemic area. This reperfusion injury is a common cause of damage in many patients [4,5,6,7]. Myocardial damage and myocardial cell death due to oxidative stress lead to a decline in cardiac function [8,9,10]. Representative changes include decreased contractility, decreased cardiac output, and structural changes such as cardiac remodeling [11,12,13,14,15]. Reducing oxidative stress in this pathological condition helps to minimize myocardial damage caused by ischemia–reperfusion injury (IRI) and offers cardioprotective benefits [16,17]. These benefits are mediated through various molecular mechanisms.

Recent studies have elucidated the role of antioxidant enzymes, such as superoxide dismutase and catalase, in mitigating oxidative stress-induced damage [18,19]. Furthermore, the activation of cellular survival pathways, including the PI3K/Akt and ERK1/2 signaling cascades [20,21,22], has been shown to confer cardioprotective effects against IRI. The modulation of mitochondrial function is another critical aspect of cardioprotection, as mitochondria are both sources and targets of reactive oxygen species during reperfusion [23,24,25]. Emerging research has also highlighted the potential of ischemic preconditioning and postconditioning strategies in reducing infarct size and preserving cardiac function. These approaches involve brief periods of ischemia and reperfusion applied either before or after the main ischemic insult, respectively, and have demonstrated promising results in experimental models and clinical trials. Additionally, pharmacological interventions targeting specific components of the oxidative stress pathway have shown potential in preclinical studies, although their translation to clinical practice requires further investigation [26,27].

The Sestrin family members also exhibit distinct regulatory mechanisms and tissue-specific expression patterns. SESN1 is ubiquitously expressed in human tissues, with the highest levels found in skeletal muscle, heart, liver, and brain. SESN2, initially identified as a hypoxia-inducible gene, is highly expressed in the kidneys, lungs, leukocytes, liver, gastrointestinal tract, and brain. SESN3, the least studied member, shows high expression in the brain, kidney, colon, small intestine, liver, and skeletal muscle [28,29,30,31]. Beyond their antioxidant functions, Sestrins play crucial roles in maintaining metabolic homeostasis. They regulate glucose homeostasis, insulin sensitivity, and lipid metabolism, primarily through their interaction with the AMPK/mTORC1 signaling pathway, where Sestrins activate AMPK and inhibit mTORC1, leading to improved metabolic outcomes. Additionally, Sestrins contribute to cellular stress responses by regulating autophagy, a critical process for cellular homeostasis. Through their interaction with the AMPK/mTOR signaling axis, Sestrins can promote autophagy induction, particularly under conditions of cellular stress or nutrient deprivation [32,33,34,35,36].

Noncoding RNAs, transcribed from DNA but not translated into proteins, are a type of functional RNA molecule. Among them, miRNAs act as post-transcriptional regulators of gene expression, and many studies have shown that changes in miRNA expression are related to various cardiovascular diseases [37,38,39]. In our previous study, we confirmed that upregulation of miR-25 suppresses the expression of SERCA2a, a master regulator of calcium signaling in cardiomyocytes [40]. This finding has significant implications for understanding the molecular mechanisms underlying cardiac dysfunction. SERCA2a plays a crucial role in maintaining calcium homeostasis in cardiomyocytes by facilitating the reuptake of calcium into the sarcoplasmic reticulum during diastole [41,42]. The reduced expression of SERCA2a has been associated with impaired cardiac contractility and relaxation, which are hallmarks of heart failure. The discovery that miR-25 regulates SERCA2a expression provides a potential therapeutic target for addressing calcium handling abnormalities in cardiovascular diseases.

Following these observations, this study also revealed a significant upregulation of miR-25 under hypoxic conditions, both in vitro and in vivo. Through a comprehensive in silico analysis utilizing three distinct miRNA target prediction algorithms, we identified SESN3, a member of the Sestrin family known for its antioxidant properties, as a potential target of miR-25.

Subsequently, we confirmed the inverse relationship between miR-25 and SESN3 expression under H/R stress conditions in vitro. Furthermore, the overexpression of SESN3 alone significantly reduced ROS-induced apoptosis. Based on these results, we delivered miR-25 TuD in vitro and in vivo models, successfully attenuating the expression of stress markers, including those associated with apoptosis and fibrosis. Notably, cardiac function was also substantially enhanced by miR-25 TuD, suggesting that its delivery may be a potential treatment for mitigating IRI-induced cardiac damage.

In summary, this study identifies SESN3 as a new target of miR-25 under oxidative stress conditions. Targeting miR-25 may serve as a promising strategy to alleviate IRI-induced cardiac dysfunction.

## 2. Materials and Methods

### 2.1. Cell Culture and Hypoxia/Reoxygenation In Vitro Model

H9c2 and HEK-293T cell lines were purchased from the American Type Culture Collection (ATCC, Manassas, VA, USA). Cells were cultured in 10 cm cell culture dishes in DMEM (CM002-050, GenDepot, Katy, TX, USA) supplemented with 10% fetal bovine serum (FBS; F0600-050, GenDepot, Katy, TX, USA) and 1% penicillin–streptomycin (CA005-010, GenDepot, Katy, TX, USA), at 37 °C with 5% CO_2_ and 95% relative humidity. Cells were subcultured every 3 days to maintain subconfluence. H9c2 cells were starved in serum-free DMEM for 24 h after reaching 70–80% confluence, then exposed to either normoxia or hypoxia conditions for an additional 24 h in a hypoxia chamber (MIC-101, Billups-Rothenberg, Del Mar, CA, USA). For hypoxic conditions, the chamber was flushed with a hypoxia gas mixture containing 1% O_2_, 5% CO_2_, and 94% N_2_, at 2 mmHg pressure for 3 min. The cells were then reoxygenated for an additional 2 h.

### 2.2. Predicted Gene Characteristics

The characteristics of the 148 predicted target genes of miR-25 were analyzed using the SRplot platform (www.bioinformatics.com (accessed on 15 November 2024)). The detailed results of this analysis are presented in Appendix A.

### 2.3. SESN3 3′-UTR Luciferase Assay

To verify that hsa-miR-25 directly binds to the SESN3 3′-UTR gene, we used a luciferase assay kit for analysis. HEK-293T cells were transfected with the plasmids described in Figure 1B using Lipofectamine 3000 (L3000015, Invitrogen, Waltham, MA, USA) transfection reagent. Measurements were made 24h after co-transfection with miRNA vectors, and luciferase (E1500, Promega, Madison, WI, USA) activity was measured using a luminescence microplate reader.

### 2.4. Measurement of Cellular ROS

ROS levels were analyzed using dichloro-dihydro-fluorescein diacetate (DCFH-DA, D6883, Sigma, Saint Louis, MO, USA) according to the manufacturer’s instructions. Briefly, a final concentration of 10 μM of DCFH-DA was added to cells upon reoxygenation, and after 2 h, cells were washed with cold PBS, observed, and images were processed using a Leica microscope (DCF295, Leica, Berlin, Germany).

### 2.5. Immunofluorescence and TdT-Mediated dUTP Nick End-Labeling (TUNEL) Staining

Cells were plated on coverslips 24 h before transfection. After experimental treatment, cells were fixed for 30 min in 4% PFA (BYLABS, Hanam, Republic of Korea) and blocked with PBS-BT (1% BSA, 0.1% Triton X-100 in PBS) for 1 h. Cells were then incubated with primary antibodies overnight at 4 °C and incubated with Alexa Fluor 594-conjugated secondary antibodies (A-11012, Thermo Fisher Scientific, Waltham, MA, USA) for 1 h at room temperature. After staining with 4,6-diamidino-2-phenylindole (DAPI; D3571, Thermo Fisher Scientific, Waltham, MA, USA) for 5 min, cells were washed three times with PBS and mounted on glass slides. TUNEL staining was performed to detect cell apoptosis in vitro according to the manufacturer’s instructions using a commercially available kit (#64936, Cell Signaling, Danvers, MA, USA).

### 2.6. Animal Care

All animal experiments in this study were performed in accordance with the Animal Care and Use Committee of Hanyang University ERICA (Approval No. HY-IACUC-24-0058, HY-IACUC-24-0059, HY-IACUC-24-0074). Eight-week-old male C57BL/6 mice were purchased from Orient Bio (Seongnam, Republic of Korea). The animals were housed in a specific pathogen-free environment at a constant ambient temperature of 22 °C and 50% humidity with a 12 h light/dark cycle. Food and water were provided ad libitum.

### 2.7. Western Blot Analysis

Cells and tissues were lysed in RIPA buffer (R4100, GenDepot, Katy, TX, USA) supplemented with protease and phosphatase inhibitors (P3300, GenDepot, Katy, TX, USA). Lysates were centrifuged at 13,000 rpm for 10 min at 4 °C. Protein concentrations in lysates were quantified using BCA assay (23225, Thermo Fisher Scientific, Waltham, MA, USA) according to the manufacturer’s instructions. Subsequently, they were transferred to nitrocellulose membranes (LC7033, GenDepot, Katy, TX, USA) via SDS-PAGE, and membranes were irradiated with primary and HRP-linked secondary antibodies and confirmed using the ECL detection system (K-12045-D50, Advansta, Menlo Park, CA, USA). Antibodies used for immunoblotting are listed in the Appendix A.

### 2.8. Quantitative Real-Time PCR

Total RNA and miRNA from cells and tissues were extracted using Hybrid-R miRNA Isolation Kit (Geneall, Seoul, Republic of Korea), following the manufacturer’s protocol. After elution, RNA purity was measured using a spectrophotometer (Multiskan SkyHigh, Thermo Fisher Scientific, Rockford, IL, USA), and only samples with a 260/280 nm ratio between 1.8 and 2.1 were used. Reverse transcription (RT) for miRNA was performed with 4 µg of total RNA in a total reaction volume of 20 µL using the Mir-X miRNA First-Strand Synthesis Kit (Takara Bio Inc., Shiga, Japan), and cDNA from total RNA was synthesized using the Tetro cDNA Synthesis Kit (BIO-65043, Bioline, Memphis, TN, USA) according to the manufacturer’s protocol. mRNA expression was analyzed using Quantstudio 1 (Applied Biosystems, Waltham, CA, USA) and the ΔΔCT method. The transcript abundance of miRNA was normalized to U6, and SESN1, SESN2, and SESN3 were normalized to 18S.

The reaction parameters were as follows for 40 cycles: 94 °C for 30 s, 60 °C for 30 s, with an initial cycle of 50 °C for 2 min followed by 95 °C for 10 min. The primers used for qRT-PCR are listed in Appendix A.

### 2.9. Myocardial Ischemia–Reperfusion Injury (IRI) Model

This was performed based on guidelines for experimental models of myocardial ischemia and infarction. Briefly, 8-week-old mice were anesthetized with ketamine (95 mg/kg) + xylazine (5 mg/kg) by intraperitoneal injection and then cannulated using a 22-gauge venous catheter. The mice received mechanical ventilation with medical oxygen. The surgery was performed on a 37 °C heating pad to prevent the body from cooling, and the left thoracotomy was performed after shaving the chest to prevent contamination during the surgery. A loose double knot was made with a suture under the left anterior descending artery, and polyethylene glycol tubing (20 gauge) was inserted and fixed, followed by ischemic injury for 30 min. After that, the knot was cut, the tube was removed, and the chest was closed in layers with sutures. In this study, 1E11vg of AAV9 was administered intravenously via tail vein.

### 2.10. Evaluation of Heart Function by Echocardiography

Functional assessment of I/R-injured hearts was performed using echocardiography. The mice were anesthetized with 2% isoflurane, and two-dimensional images and M-mode tracings were recorded in the short axis at the level of the left ventricular papillary muscles using a 14.0 MHz transducer to determine fractional shortening ratio and ventricular dimensions (Vivid S60N; GE HealthCare, Chicago, IL, USA). EF and FS, indices of LV systolic function, were calculated using the following equations, respectively:EF (%) = 100 × [(LVIDd^2^ − LVIDs^2^)/LVIDd^2^]FS (%) = 100 × [(LVIDd − LVIDs)/LVIDd]

### 2.11. Statistical Analysis

Statistical analyses for all experiments were performed using Prism version 9.5 (GraphPad Software, Inc., La Jolla, CA, USA). Data are representative of independent experiments and are expressed as the mean ± standard error of the mean (SEM). One-way ANOVA and Student’s *t*-test were used to compare the groups, and * *p* < 0.05, ** *p* < 0.01, and *** *p* < 0.001 were considered statistically significant at different levels.

## 3. Results

### 3.1. SESN3 Is a Newly Identified Target of miR-25 as an ROS Scavenger

IRI induces severe oxidative stress in the heart [43,44,45], subsequently leading to myocardial cell damage and cardiac dysfunction. Recently, we observed a significant increase in the expression of miR-25 under hypoxia/reoxygenation (H/R) conditions, and pretreatment with miR-25 Tough Decoy (TuD) substantially reduced ROS generation. Therefore, we hypothesized that miR-25 may regulate potential antioxidant molecules. To identify novel targets of miR-25 involved in the regulation of oxidative damage, we conducted in silico analyses utilizing three databases: Target Miner (May 2012 version), TargetScan7 (TargetScanHuman 7.0), and miRDB (https://mirdb.org/ (accessed on 18 May 2022)). From a total of 148 candidate genes identified through these databases, we selected SESN3 for further investigation (Figure 1A, Appendix A) due to its antioxidant properties. Notably, the 3′-UTR of the SESN3 mRNA exhibited high conservation across species.

To verify the direct interaction between miR-25 and SESN3, we performed a luciferase reporter assay using a construct containing the SESN3 3′-UTR sequence. The results demonstrated a dose-dependent decrease in luciferase activity with increasing concentrations of pre-miR-25, suggesting that miR-25 directly targets the 3′-UTR region of SESN3 (Figure 1B). To validate these findings, we employed an in vitro hypoxia/reoxygenation (H/R) system using H9c2 myoblasts and evaluated the effect of inhibiting miR-25 on the SESN3 expression and ROS generation. Under H/R conditions, we first confirmed that miR-25 was highly expressed (Figure 1C). Importantly, pretreatment with the miR-25 TuD inhibitor under H/R conditions not only normalized the SESN3 expression (Figure 1C) but also significantly reduced ROS generation, as shown in Figure 1D. Collectively, these results provide compelling evidence that H/R-induced ROS generation is significantly reduced by the restoration of SESN3 expression through miR-25 inhibition.

### 3.2. SESN3 Overexpression Ameliorates ROS-Induced Apoptosis In Vitro

The SESN family consists of three isoforms: SESN1, SESN2, and SESN3. Most of the literature has focused on SESN1 and SESN2, likely due to their relatively higher expression levels [46,47,48,49]. In contrast, the antioxidant effects of SESN3, particularly in the heart, have not been fully elucidated. Therefore, we first characterized SESN3 activity to determine whether it can also scavenge ROS under hypoxia/reoxygenation (H/R) conditions. To this end, we evaluated the effect of SESN3 overexpression by measuring immunofluorescence against ROS and cell death under the H/R condition.

First, we observed that the apoptotic markers, cleaved PARP and cleaved caspase-9, significantly increased in the H/R group; however, these markers were substantially reduced with SESN3 overexpression (Figure 2A). These findings suggest that SESN3 effectively mitigates ROS-induced cell death (Figure 2B,C).

Based on these results, we further investigated the impact of miR-25 on SESN3 expression by treating cells with pre-miR-25 and performing immunofluorescence staining for SESN3. As shown in Appendix A, the SESN3 expression was dramatically decreased in the pre-miR-25 transfected group. In summary, we confirmed that SESN3 alone fully mitigates ROS-induced apoptosis, and its expression can be negatively regulated by miR-25.

### 3.3. miR-25 TuD Delivery Attenuates Hypoxia/Reoxygenation-Induced Damage in H9c2 Cardiomyoblasts

Based on previous results, we next measured the expression of three forms of miR-25 (pri-, pre-, and mature forms) and three SESN isoforms following miR-25 TuD delivery to assess its specificity for SESN3. First, all three forms of miR-25 were substantially increased under H/R conditions, but miR-25 TuD delivery dramatically normalized their expression (Figure 3B). Under the same conditions, we examined the expression of all three SESN isoforms: SESN1, SESN2, and SESN3. SESN1 and SESN2 were significantly upregulated under H/R conditions, consistent with previous reports [49]. In contrast, the SESN3 expression was significantly decreased. To our surprise, miR-25 TuD delivery significantly increased the SESN3 expression while both SESN1 and SESN2 were downregulated (Figure 3C). These results suggest that normalizing SESN3 expression is sufficient to reduce ROS-induced damage, indicating that SESN1 and SESN2 levels could also normalize as a consequence.

Oxidative stress generally induces cell apoptosis, which, in many cases, exacerbates cardiac fibrosis [50]. Therefore, we further investigated the effect of miR-25 inhibition in both apoptosis and fibrosis. As shown in Figure 3D, the restoration of SESN3 expression through miR-25 TuD delivery remarkably decreased the levels of apoptotic markers, including cleaved PARP and cleaved caspase-9, while increasing the expression of the anti-apoptotic marker Bcl-XL. Furthermore, we observed a reduction in the expression of fibrosis markers, such as TGF-β and fibronectin (Figure 3D), and Collagen I/III (Appendix A). Taken together, these results suggest that miR-25 inhibition via TuD delivery not only modulates the expression of the SESN family but also attenuates apoptosis and fibrosis through the regulation of oxidative stress.

### 3.4. miR-25 TuD Delivery Reduces ROS-Mediated Damage in a Cardiac IRI Model

Our results suggest that miR-25 inhibition successfully reduced ROS generation, apoptosis, and fibrosis through the restoration of SESN3 expression in vitro. To validate these findings and explore their clinical relevance, we extended our investigation to an in vivo mouse model of cardiac IRI, with or without miR-25 TuD delivery.

For this purpose, we utilized AAV9-EGFP (control) and AAV9-miR-25 TuD, administered via tail vein injection to 8-week-old mice, according to the experimental scheme (Figure 4A). Prior to evaluating cardiac function, we analyzed the biodistribution of AAV9-GFP. The dissection of five major organs (brain, heart, lungs, liver, and kidneys) revealed that the heart exhibited the second highest level of expression (Appendix A), indicating that AAV9 has significant adaptability for cardiac tissue [51,52,53]. We then confirmed that AAV9-miR-25 TuD delivery effectively reduced the levels of primary miR-25, precursor miR-25, and mature miR-25-3p in the mouse heart (Figure 4B). This reduction in miR-25 expression is crucial for the success of our subsequent experiments.

We then examined the expression of the SESN isoforms, focusing particularly on SESN3, a novel target of miR-25. As shown in Figure 4C, SESN3 mRNA expression was restored following AAV9-miR-25 TuD delivery, while SESN1 and SESN2 were substantially reduced, consistent with our in vitro data. Next, we investigated the expression of key proteins related to apoptosis and fibrosis in the cardiac samples from these mice. As in in vitro conditions, the SESN3 expression was significantly increased. Additionally, cleaved caspase-9, a key mediator of apoptosis, was highly elevated but effectively mitigated by miR-25 TuD delivery (Figure 4D). In contrast, the anti-apoptotic marker Bcl-XL was substantially upregulated, indicating a shift towards cell survival and away from apoptotic pathways.

Furthermore, we observed a reduction in the TGF-β and fibronectin levels, both of which are associated with fibrotic processes, along with significant decreases in Collagen Type I and Type III (Figure 4D, Appendix A). These findings indicate that miR-25 inhibition not only prevents cell apoptosis but also attenuates fibrotic remodeling in the injured myocardium.

In summary, our data demonstrate that miR-25 inhibition using miR-25 TuD is sufficient to alleviate oxidative stress in cardiac IRI. The restoration of SESN3 expression, along with the modulation of apoptotic and fibrotic markers, appears to be a key mechanism in preventing cell death and pathological remodeling.

### 3.5. miR-25 TuD Delivery Mitigates IRI-Induced Cardiac Dysfunction

To assess the impact of miR-25 TuD delivery on cardiac function, we performed weekly echocardiographic evaluations (Figure 5A). In the IRI model, the heart weight-to-body weight ratio was increased compared to that in the sham-operated mice, while the mice treated with AAV9 miR-25 TuD exhibited a ratio similar to that of the sham group (Figure 5B). Ventricular ejection fraction (EF) and fractional shortening (FS) were significantly higher in the AAV9 miR-25 TuD-treated group than in the IRI group (Figure 5C). The decreased IVSd and IVSs values observed in the IRI model suggest a loss of contractility due to ischemic injury. Notably, both values were preserved in the mice treated with AAV9 miR-25 TuD (Figure 5C).

Taken together, these results suggest that miR-25 inhibition using miR-25 TuD can effectively prevent ROS-induced cardiac dysfunction in vivo.

### 3.6. Knockdown of SESN3 Neutralizes the Effect of miR-25 TuD Delivery

Finally, we conducted additional experiments to verify whether the beneficial effects of miR-25 inhibition are mediated specifically through the regulation of SESN3 expression under H/R conditions. To this end, we used shSESN3 to knock down the SESN3 expression in combination with miR-25 TuD delivery. As shown in Figure 6A, the SESN3 expression was completely absent following SESN3 knockdown, even with miR-25 TuD treatment (Figure 6A, lanes 6–7). Furthermore, the beneficial effects of miR-25 TuD treatment were completely abrogated (Figure 6A). These findings were also substantiated at the mRNA level (Figure 6B), providing strong evidence that SESN3 is indeed a direct target of miR-25.

Consequently, these results highlight the specificity of miR-25 in regulating SESN3 expression and demonstrate that the anti-apoptotic and anti-fibrotic effects of miR-25 TuD are mediated through SESN3 by reducing oxidative damage in the heart.

## 4. Discussion

The treatment of acute myocardial ischemia presents a paradoxical challenge in modern cardiology. While early and successful reperfusion remains the most effective strategy for salvaging ischemic myocardium, it can also exacerbate myocardial damage through a phenomenon known as ischemia–reperfusion injury (IRI) [4,5]. This injury manifests through various mechanisms, including endothelial dysfunction, myocardial cell necrosis, alterations in calcium handling, metabolic disturbances, and inflammation [54]. A significant contributor to IRI is the production of reactive oxygen species (ROS) and intracellular Ca^2+^ overload, both of which play central roles in myocardial damage.

Recent studies have shown that certain miRNAs play crucial roles in regulating oxidative stress responses in cardiovascular diseases. For instance, miR-15b, miR-29, and miR-222 have been shown to downregulate key antioxidant factors such as SIRT4, PGC-1α, and SOD, respectively, thereby disrupting cellular defenses against oxidative stress [55,56,57]. These findings underscore the potential of miRNAs as both biomarkers and therapeutic targets in cardiovascular pathologies. Interestingly, studies have shown contradictory roles for miR-25: while one study reported that increased miR-25 expression exacerbated ROS-induced cell damage [58], another study suggested that miR-25 upregulation protects cardiomyocytes from oxidative stress [59]. These opposing findings may be attributed to differences in experimental models, specific stressors, or the timing and duration of miR-25 modulation.

Our recent studies revealed that miR-25 is highly upregulated in the IRI model, and its overexpression exacerbates cardiac damage and impaired function. Thus, we hypothesized that miR-25 may have an additional target involved in regulating ROS generation. To explore this, we conducted in silico screening and identified SESN3 as a novel target of miR-25 in the context of ROS regulation.

The Sestrin family of proteins performs critical functions, such as protecting against oxidative stress [60], regulating mTORC1 signaling [61], and promoting autophagy [62]. Sestrin3, in particular, has gained attention for its diverse roles in various tissues. For example, in the liver, Sestrin3 has been shown to protect against metabolic stress and insulin resistance [31], while also modulating FOXO3-mediated ROS regulation [63]. However, its role is not universally protective across all tissues. In neurological diseases, Sestrin3 has been implicated in the promotion of epileptic seizures [64], and in a mouse model of colitis, increased Sestrin3 expression activated macrophages and promoted the production of inflammatory cytokines [65]. This highlights the context-dependent nature of Sestrin3 function and underscores the importance of carefully considering its therapeutic targeting.

The role of Sestrin3 in cardiovascular diseases, particularly in cardiac IRI, remains underexplored. While Sestrin2 has been shown to have protective roles in cardiac IRI, its interaction with Sestrin3 in this context has not been fully elucidated. Therefore, we performed experiments using an miR-25 TuD inhibitor to test our hypothesis. As expected, when the ROS levels increased under H/R conditions, the Sestrin3 expression was suppressed by pre-miR-25. However, miR-25 TuD delivery restored Sestrin3 expression (Figure 1C,D). Additionally, Sestrin3 overexpression alleviated ROS-induced apoptosis (Figure 2A,B). Further experiments confirmed that miR-25 TuD delivery reduced apoptosis and fibrosis (Figure 3D and Figure 4C), consistent with previous findings that Sestrin3 inhibits TGF-β signaling to prevent fibrosis [66].

While this study provides strong evidence that miR-25 directly targets SESN3 and modulates oxidative stress-induced apoptosis and cardiac fibrosis, several areas require further investigation. First, the precise molecular mechanisms by which SESN3 overexpression mitigates ROS-induced apoptosis remain unclear. Elucidating these downstream signaling pathways could provide deeper insights into the protective effects of SESN3. Second, although we demonstrated that miR-25 regulates SESN3 expression in both H/R and IRI models, further research is needed to explore the regulation of other SESN isoforms under miR-25 TuD treatment. Lastly, while SESN3 is a confirmed target of miR-25, it is also important to investigate whether other miR-25 targets play significant roles in modulating oxidative stress and apoptosis.

Although this study demonstrates that miR-25 directly targets SESN3 and modulates oxidative stress-induced apoptosis and cardiac fibrosis, the timing of potential treatment strategies must be considered. Previous studies have shown that members of the miR-15 family confer cardioprotective effects when administered during ischemia–reperfusion injury [67]. Additionally, the inhibition of miR-192 and miR-128 prior to ischemic injury has been shown to significantly reduce infarct size, further highlighting their cardioprotective potential [68,69]. Moreover, reperfusion is most effective when performed within 90 min of ischemia onset [70,71]. Taken together, these findings suggest that therapeutic approaches targeting the miR-25/SESN3 axis would be most beneficial as a preventive measure in high-risk patients or as an immediate intervention following myocardial infarction. Future studies should focus on optimizing the timing and delivery of these treatments to maximize their cardioprotective potential.

## 5. Conclusions

Restoring blood flow is essential for treating acute myocardial ischemia, but reperfusion can paradoxically exacerbate myocardial damage through mechanisms such as ROS accumulation, calcium overload, and inflammation. In this study, we confirmed that miR-25 expression is upregulated in ischemia–reperfusion injury (IRI), and through in silico methods, we identified Sestrin3 (SESN3) as a novel target of miR-25. Additionally, we observed that using miR-25 Tough Decoy (TuD), a previously validated inhibitor of miR-25, restored SESN3 expression. This restoration of SESN3 expression resulted in reduced ROS-induced damage and improved cardiac function in both in vitro and in vivo models. In conclusion, these findings suggest that targeting the miR-25-SESN3 axis could be a promising therapeutic strategy for alleviating I/R injury.

## Figures and Tables

**Figure 1 antioxidants-14-00061-f001:**
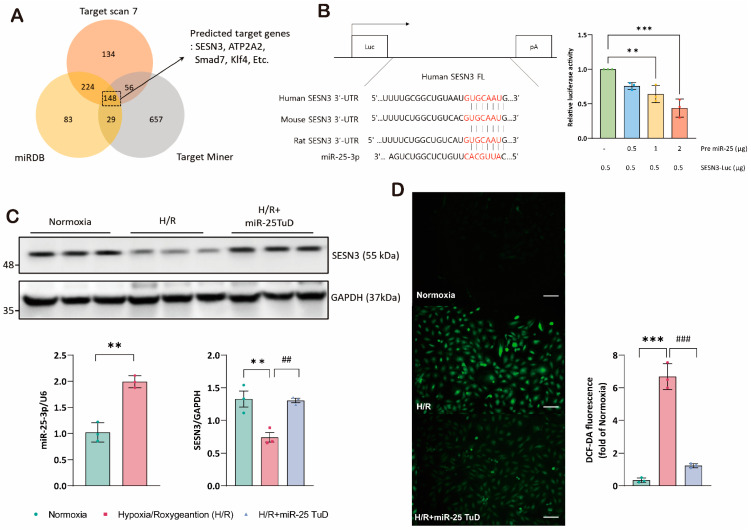
SESN3 is newly identified target of miR-25 as an ROS scavenger. (**A**) In silico identification of potential target genes for miR-25-3p. (**B**) The putative miR-25-3p binding site in the 3′-UTR region of the SESN3 mRNA was predicted using TargetScan. Pre-miR-25 and SESN3-Luc plasmids were transfected into HEK-293T cells. SESN3-Luciferase activity was subsequently analyzed in response to increasing pre-miR-25 transfection levels. (**C**) Protein expression of SESN3 was analyzed in H9c2 cells transfected with miR-25 TuD plasmids. (**D**) Representative fluorescence microscopy images of H9c2 cells stained with DCFDA to detect intracellular ROS levels. A scale bar indicates 50 μm respectively. (** *p* < 0.01, *** *p* < 0.001 vs. Normoxia, ## *p* < 0.01, ### *p* < 0.001 vs. H/R).

**Figure 2 antioxidants-14-00061-f002:**
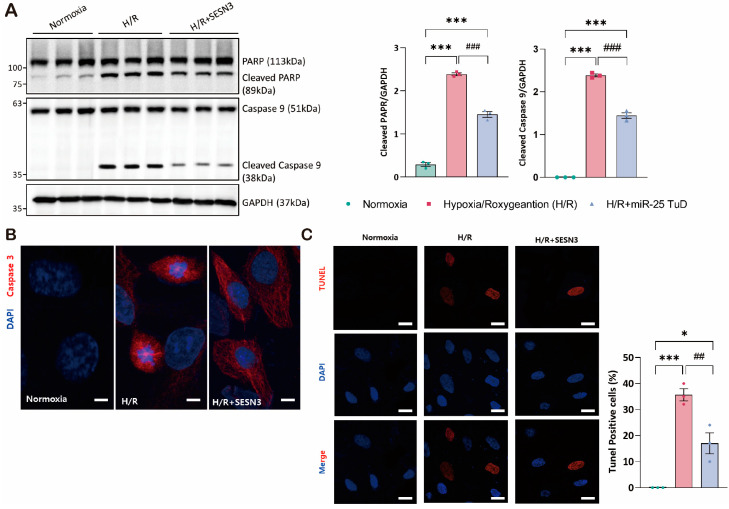
SESN3 overexpression ameliorates ROS-induced apoptosis in vitro. (**A**) Western blot analysis of SESN3, Caspase 9, and cleaved caspase 9 protein levels in myoblasts under Normoxia, HR, and HR+SESN3 overexpression conditions. GAPDH served as a loading control. Representative blots from three independent experiments are shown. (**B**) Representative fluorescence microscopy images of myoblasts stained for active caspase 3 (red). Nuclei were counterstained with DAPI (blue). The scale bar indicates 25 μm. (**C**) TUNEL assay for detection of apoptotic cells. Representative images showing TUNEL-positive nuclei (red) and total nuclei stained with DAPI (blue). The scale bar indicates 50 μm. (* *p* < 0.05, *** *p* < 0.001 vs. Normoxia, ## *p* < 0.01, ### *p* < 0.001 vs. H/R).

**Figure 3 antioxidants-14-00061-f003:**
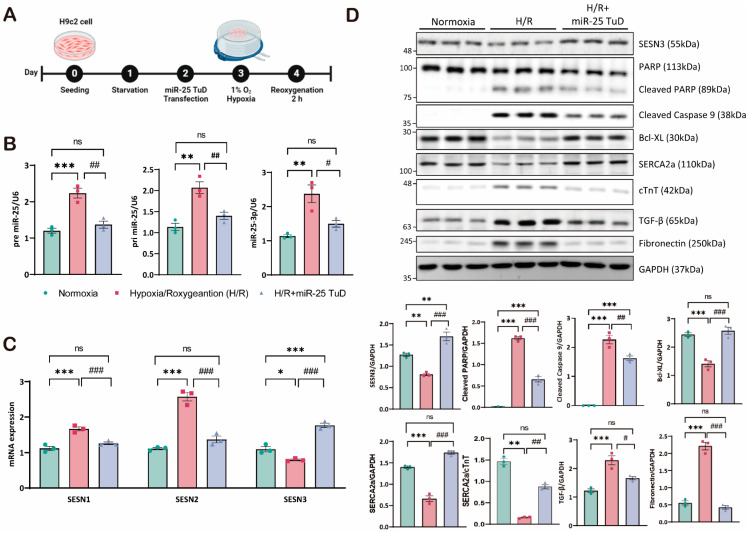
miR-25 TuD delivery attenuates hypoxia/reoxygenation-induced damage in H9c2 cardiomyoblasts. (**A**) Experimental design and procedures. (**B**,**C**) The expression levels of miR-25 (primary, precursor, and mature) and the expression levels of the SESN family were analyzed by qRT-PCR. (**D**) Immunoblotting analysis and quantification of protein levels in H9c2 cells subjected to hypoxia/reoxygenation (H/R) injury (* *p* < 0.05, ** *p* < 0.01, *** *p* < 0.001 vs. Normoxia, # *p* < 0.05, ## *p* < 0.01, ### *p* < 0.001 vs. H/R, ns: not significant).

**Figure 4 antioxidants-14-00061-f004:**
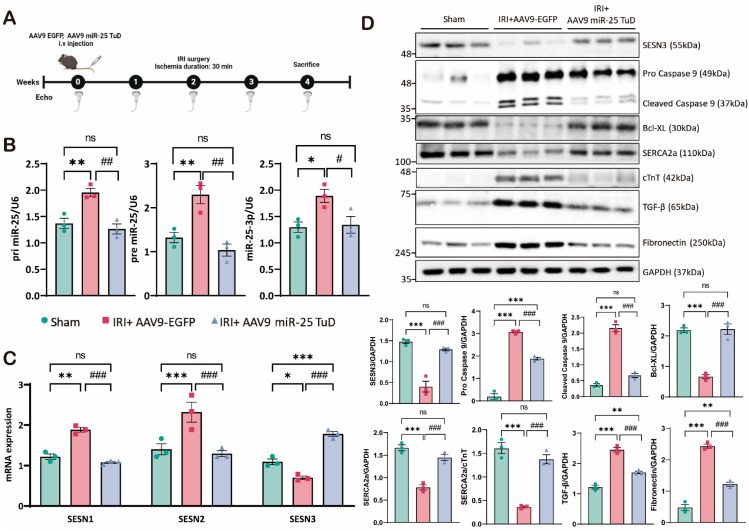
miR-25 TuD delivery reduces ROS-mediated damage in a cardiac IRI model. (**A**) Experimental scheme for the pretreatment of miR-25 TuD in the ischemia–reperfusion injury model. (**B**) The expression levels of different forms of endogenous miR-25 (primary, precursor, and mature) were evaluated to confirm the long-term effect of AAV9 miR-25 TuD. (**C**) SESNS family expression levels were analyzed using qRT-PCR. (**D**) Immunoblotting analysis and quantification of protein levels in the ischemia–reperfusion injury (IRI) mouse heart (* *p* < 0.05, ** *p* < 0.01, *** *p* < 0.001 vs. Sham, # *p* < 0.05, ## *p* < 0.01, ### *p* < 0.001 vs. IRI+AAV9-EGFP, ns: not significant).

**Figure 5 antioxidants-14-00061-f005:**
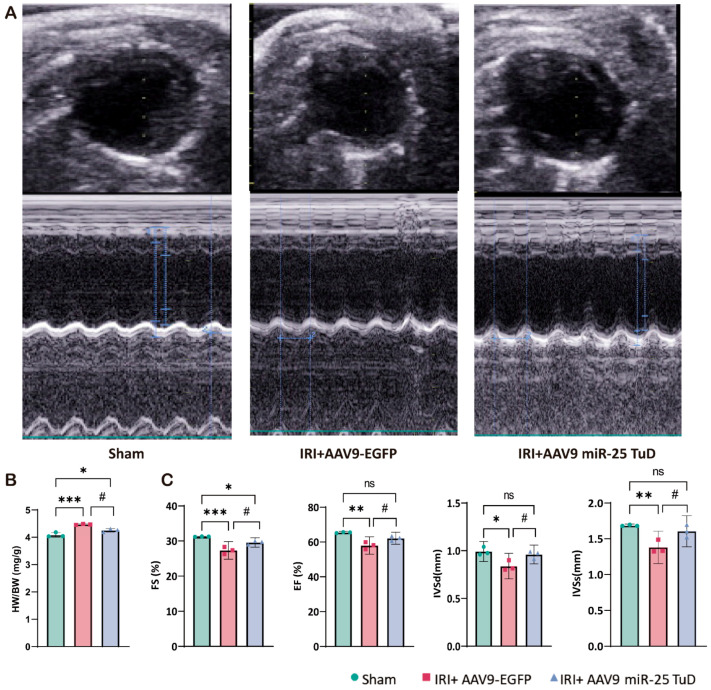
miR-25 TuD delivery mitigates IRI-induced cardiac dysfunction. (**A**) Representative left ventricular M-mode echocardiographic images of the sham, AAV9-EGFP, and AAV9 miR-25 TuD-transfected groups were obtained before sacrifice. (**B**) Quantitative results of heart weight/body weight (HW/BW) 2 weeks post-I/R surgery. (**C**) Left ventricular functional and remodeling parameters, including FS, EF, IVSd, and IVSs, are presented (* *p* < 0.05, ** *p* < 0.01, *** *p* < 0.001 vs. Sham, # *p* < 0.05 vs. IRI+AAV9-EGFP, ns: not significant).

**Figure 6 antioxidants-14-00061-f006:**
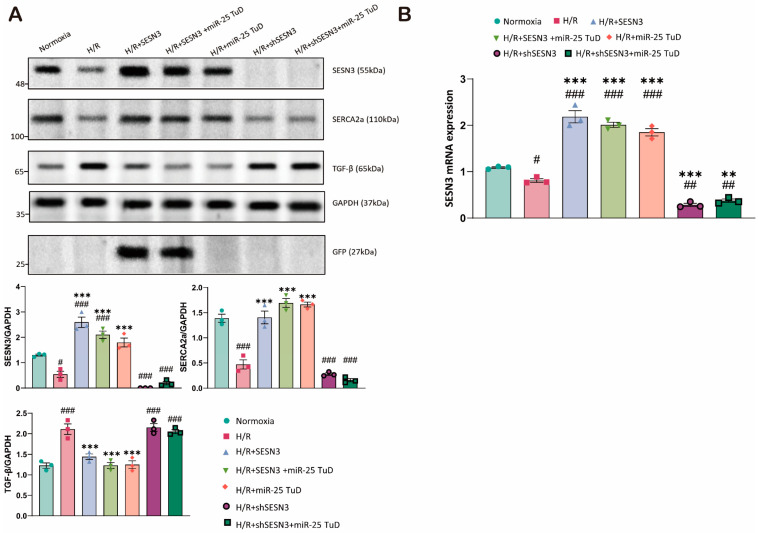
Knockdown of SESN3 neutralizes the effect of miR-25 TuD delivery. (**A**) SESN3 overexpression, miR-25 TuD treatments, and SESN3 knockdown were applied either individually or in various combinations in H9c2 cells under H/R conditions. Immunoblotting was then performed using anti-SERCA2a, TGF-β, and GFP antibodies. (**B**) SESN3 mRNA expression was measured in H9c2 cells under hypoxia/reoxygenation (H/R) conditions (** *p* < 0.01, *** *p* < 0.001 vs. Normoxia, # *p* < 0.05, ## *p* < 0.01, ### *p* < 0.001 vs. H/R).

## Data Availability

Data is contained within the article or Appendix A.

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
