# Peer review of "Restoration of Sestrin 3 Expression Mitigates Cardiac Oxidative Damage in Ischemia–Reperfusion Injury Model"

_antioxidants, 2025, doi:10.3390/antiox14010061_

Round 1

Reviewer 1 Report (Previous Reviewer 1)

A few major questions remain to accept this manuscript.

  • No supplemental Table 1
  • No supplemental Figure 1
  • Few reagents information in the methods section
  • Results section: 3.1, lines 197-200, reference is missing
  • Figure 2C, the graphic is different from the other. Please maintain consistency, indicating the n.
  • Collagens should be evaluated in vitro and in vivo. The authors used a myoblast cell line that presented plasticity, the results indicating the expression of fibronectin can indicate switching of myoblast to myofibroblast. Additionally, the cardiomyocyte expression should also be evaluated because the decrease in SERCA2a can be caused by fewer cardiomyocytes in the sample. The SERCA2a expression also must be normalized by any cardiac marker, like cTNT or cTNI.

A few major questions remain to accept this manuscript.

  • No supplemental Table 1
  • No supplemental Figure 1
  • Few reagents information in the methods section
  • Results section: 3.1, lines 197-200, reference is missing
  • Figure 2C, the graphic is different from the other. Please maintain consistency, indicating the n.
  • Collagens should be evaluated in vitro and in vivo. The authors used a myoblast cell line that presented plasticity, the results indicating the expression of fibronectin can indicate switching of myoblast to myofibroblast. Additionally, the cardiomyocyte expression should also be evaluated because the decrease in SERCA2a can be caused by fewer cardiomyocytes in the sample. The SERCA2a expression also must be normalized by any cardiac marker, like cTNT or cTNI.

Author Response

  1. No supplemental Table 1
    Response:

We apologize for the oversight in not including Supplemental Table 1 in last submission. It will be properly attached in the second revised manuscript.

  1. No supplemental Figure 1

Response:

In the previously revised manuscript, we had already provided supplemental data, including this figure. However, we will include it again in the second revised submission to ensure clarity.

  1. Few reagents information in the methods section.

Response:

Full information about all reagents has been provided in the second revised manuscript.

  1. Results section: 3.1, lines 197-200, reference is missing

Response:

We deeply apologize for the oversight in missing references. They will be included appropriately (line 199).

  1. Figure 2C, the graphic is different from the other. Please maintain consistency, indicating the n.

Response:

Thank you for your thoughtful comments. We will make the necessary modifications to all figures to ensure consistency.

  1. Collagens should be evaluated in vitro and in vivo. The authors used a myoblast cell line that presented plasticity, the results indicating the expression of fibronectin can indicate switching of myoblast to myofibroblast. Additionally, the cardiomyocyte expression should also be evaluated because the decrease in SERCA2a can be caused by fewer cardiomyocytes in the sample. The SERCA2a expression also must be normalized by any cardiac marker, like cTNT or cTNI.

Response:

Thank you for your suggestions. We agree with the reviewer’s comment and have therefore reperformed the western blot for cTnT expression, which was used as an additional normalization control for SERCA2a expression both in vitro and in vivo (Figure 3D and 4D). Furthermore, additional collagen expression data were measured by qRT-PCR in the in vitro model and are now included in Supplementary Figure 4.

Reviewer 2 Report (Previous Reviewer 2)

I believe authors addressed all my concerns.  

I believe authors addressed all my concerns. 

Author Response

Thank you for your thoughtful consideration!!

Round 2

Reviewer 1 Report (Previous Reviewer 1)

The authors performed all requested editions to the manuscript. 

No additional comment.

This manuscript is a resubmission of an earlier submission. The following is a list of the peer review reports and author responses from that submission.

Round 1

Reviewer 1 Report

Understanding the mechanisms of ischemia-reperfusion injury (IRI) in the cardiovascular system is essential for proposing new therapeutic approaches. Park et al. hypothesized that miR25, via SESTRIN3, is a good target for decreasing oxidative damage. The manuscript is well-written, and a few raised concerns must be addressed before it is considered for publication.

Major revision:

  • It is well known that miRs from human and animal models usually didn’t present the same targets and effects. Did the authors perform any miR25 vs SESTRIN3 experiments and observe similar results to the ones observed in this study?
  • The cell line used in this study is a commercial rat myoblast, which does not have all the features of a cardiomyocyte. It is hard to assume that the effect observed is on cardiomyocytes when the cells present plasticity under stress. Could the authors provide any data demonstrating that the cells under stress (H/R, miR-delivering, and si-delivering) maintained the cardiac phenotype in vitro? This information is also essential when the authors compare the in vitro data of TGFb expression in Fig 3 and SERCA2a and TGFb expression in Fig 6.
  • It would be interesting if the authors created a supplementary file with the 148 predicted genes and their function/characteristics. This enhances the relevance of the findings and may be helpful for further studies.
  • Did the authors perform an organ-delivery assay? For example, how many EGFP was detected in the heart? And how long was the EGFP detected?
  • When the authors performed the injury (week 2), was the miR25 still detected in the animal system?
  • The authors should demonstrate any collagen expression to affirm that the miR25-TUD attenuated fibrosis. Mainly Col I and Col III.
  • Why were the SESN1 and 2 substantially reduced if there is no statistical difference between Sham and IRI+mir25-TUD? Is it a protective effect over the process, or does the injury happen, but the recovery was effective?

Minor:

  • A few manufacturers’ information is missing in the methods section. Please revise it.
  • Section 2.1. Please describe the media used when the cells went into starvation.
  • In the results section (3.1), lines 201-204, is the sentence already published or “data not shown”? Please include the reference or information data not shown in the text.
  • Figure 2A is missing from the text.

Reviewer 2 Report

Authors presented an interesting study on the role of miR-25 and SESN3 in ischemia-reperfusion injury. However there are two major problems.

1. The presented echocardiography data (Fig. 5) are of very low quality and improperly scored: (1) too fuzzy, (2) the acquisition window is not properly focused on the heart, (3) the wall motion on the images is too small to produce FS and EF shown in the quantification panel (5C), and (4) error bars in the figure are not ±SEM as described in the Methods (if it were ±SEM, 1-2 data points should have been outside of the whiskers).

2. The authors need to mention in the Discussion that any treatment strategy involving mir-25/SESN3 should pre-empt MI episode and most likely is useful only for instances of timely reperfusion (<90 min) post MI.

line 265: Fig. 3D needs to be described in the Results section.

line 274: Since 4D is the first panel to be referenced in the Fig. 4, it should be called 4A and the figure updated accordingly.

line 346: check error bars of Fig. 6. They do not look like ±SEM.

Suppl. Fig. 1: The legend is a bit confusing. Please, label clearly with wording which images are before and after the transfection